# Retrospective Analysis of Severe Dengue by Dengue Virus Serotypes in a Population with Social Security, Mexico 2023

**DOI:** 10.3390/v16050769

**Published:** 2024-05-13

**Authors:** Porfirio Felipe Hernández Bautista, David Alejandro Cabrera Gaytán, Clara Esperanza Santacruz Tinoco, Alfonso Vallejos Parás, Julio Elias Alvarado Yaah, Bernardo Martínez Miguel, Yu Mei Anguiano Hernández, Lumumba Arriaga Nieto, Alejandro Moctezuma Paz, Leticia Jaimes Betancourt, Yadira Pérez Andrade, Oscar Cruz Orozco, Gabriel Valle Alvarado, Mónica Grisel Rivera Mahey

**Affiliations:** 1Coordinación de Calidad de Insumos y Laboratorios Especializados, Instituto Mexicano del Seguro Social, Ciudad de México 07760, Mexico; porfirio.hernandez@imss.gob.mx (P.F.H.B.); clara.santacruz@imss.gob.mx (C.E.S.T.); julio.alvaradoy@imss.gob.mx (J.E.A.Y.); bernardo.martinezm@imss.gob.mx (B.M.M.); yu.anguiano@imss.gob.mx (Y.M.A.H.); 2Coordinación de Vigilancia Epidemiológica, Instituto Mexicano del Seguro Social, Ciudad de México 03100, Mexico; lumumba.arriaga@imss.gob.mx (L.A.N.); yadira.perezan@imss.gob.mx (Y.P.A.); oscar.cruzo@imss.gob.mx (O.C.O.); gabriel.valle@imss.gob.mx (G.V.A.); monica.riverama@imss.gob.mx (M.G.R.M.); 3Coordinación de Investigación en Salud, Instituto Mexicano del Seguro Social, Ciudad de México 06720, Mexico; alejandro.moctezuma@imss.gob.mx; 4Unidad de Medicina Familiar No. 7, Instituto Mexicano del Seguro Social, Ciudad de México 14370, Mexico; leticia.jaimesb@imss.gob.mx

**Keywords:** dengue, serotypes, RT-qPCR, severe dengue

## Abstract

Background: Risk factors for severe dengue manifestations have been attributed to various factors, including specific serotypes, sex, and age. Mexico has seen the re-emergence of DENV-3, which has not circulated in a decade. Objective: To describe dengue serotypes by age, sex, and their association with disease severity in dengue-positive serum samples from epidemiological surveillance system units. Materials and Methods: A descriptive analysis was conducted to evaluate the frequency of dengue severity by sex, age, disease quarter, geographical location, and dengue virus serotypes. The study was conducted using laboratory samples from confirmed dengue cases through RT-qPCR from the epidemiological surveillance laboratory network of the Mexican Social Security Institute, Mexico. Simple frequencies and proportions were calculated using the z-test for proportional differences between groups. Bivariate analysis with adjusted Chi2 was performed, and binary logistic regression models were constructed using the forward Wald method considering the model’s predictive capacity. The measure of association was the odds ratio, with 95% confidence intervals. Statistical significance was set to an alpha level of <0.05. Results: In 2023, 10,441 samples were processed for dengue RT-qPCR at the IMSS, with a predominance of serotype DENV-3 (64.4%). The samples were mostly from women (52.0%) and outpatient cases (63.3%). The distribution of dengue severity showed significant variations by age, with a lower proportion of severe cases in young children and a higher proportion in the 5- to 14-year-old group. Hospitalizations increased significantly with severity. Warm regions had more cases overall and severity. Cases were most frequent from July to September. While DENV-2 was associated with severity, DENV-4 was not. Binary regression identified higher risk in women, age extremes, and DENV-2, with an overall predictive model of 58.5%. Conclusions: Women, age groups at the extremes of life, and the DENV-2 serotype presented severe risk of dengue in a population with social security in Mexico during 2023.

## 1. Introduction

Dengue virus (DENV) infection is currently the most important mosquito-borne disease worldwide. It is predominantly distributed in tropical and subtropical regions [1]. DENV is a single-stranded RNA virus belonging to the Flaviviridae family and the Flavivirus genus [2]. The virus comprises four known serotypes (DENV 1, DENV 2, DENV 3, and DENV 4), each with distinct antigenic properties, present in the Americas [3]. In 2023, dengue reached a historical peak with over five million cases and more than 5000 deaths globally, with approximately 80% of cases reported in the Americas region [4,5]. The clinical spectrum of dengue varies widely, from asymptomatic infection to severe and potentially fatal illness [6]. Symptomatic cases manifest with a range of clinical presentations, from mild febrile illness to severe disease typically characterized by systemic vascular leakage syndrome, plasma leakage, hemostatic abnormalities, and thrombocytopenia [7]. In 2024, the Pan American Health Organization issued an epidemiological alert due to sustained circulation, primarily driven by DENV-3, which had not circulated in some areas for several years [8]. Mexico is currently considered hyperendemic for dengue, with all four dengue serotypes continuously circulating in the same geographical location [9]. Additionally, Mexico has a high incidence rate, with an increasing impact of the disease in recent decades [10]. According to the Ministry of Health, Mexico confirmed 54,406 laboratory-confirmed dengue cases in 2023, with 46.7% classified as severe dengue (severe dengue (3.1%) and dengue with warning signs (43.6%)), setting a historical record for dengue cases in Mexico [11]. Serotyping of 21,863 confirmed dengue cases revealed DENV3 (59.3%), DENV2 (21.8%), DENV1 (16.8%), and DENV4 (2.1%) [11]. Clinical classification of dengue in Mexico follows WHO guidelines, categorizing it into three stages: dengue without warning signs (non-severe dengue), dengue with warning signs, and severe dengue [12]. Risk factors for severe dengue manifestations have been attributed to various factors, including secondary versus primary infection, specific serotypes or genotypes, sex, and age. While much attention has been paid to factors causing severe and hemorrhagic disease, fewer studies have compared differences in specific clinical manifestations according to DENV serotype [13]. The objective of this retrospective cross-sectional observational study was to estimate and compare the severity of dengue illness according to DENV serotypes and other demographic characteristics, such as sex, age, climate region, and infection quarter, among individuals with social security in Mexico during 2023.

## 2. Materials and Methods

An observational, cross-sectional, retrospective study was conducted based on the clinical classification of laboratory-confirmed dengue patients and dengue virus serotyping.

Dengue virus serotyping was performed using molecular biology techniques, specifically real-time reverse transcription polymerase chain reaction (RT-qPCR), on blood serum samples from confirmed dengue cases reported by a laboratory-based epidemiological surveillance system. No antibody techniques were used.

All laboratory samples were collected within 5 days of patients’ clinical symptoms onset. Samples from patients with symptoms onset between 1 January and 31 December 2023 were included.

All laboratory samples were analyzed by the Epidemiology Laboratories Network of the Mexican Social Security Institute (IMSS), accredited for dengue diagnosis by the National Institute of Diagnostic and Epidemiological Reference of the Mexican Ministry of Health.

Based on WHO guidelines, each patient was clinically classified using the operational case definition of probable dengue cases used in Mexico. Therefore, all patients included in the study met the epidemiological operational case definition of probable dengue and were notified through epidemiological surveillance.

The clinical classification was defined as follows:(a)Probable non-severe dengue: Persons of any age residing or originating, within 14 days of symptoms onset, from a region with disease transmission, presenting fever and signs/symptoms from two or more of the following groups: Group 1: nausea and/or vomiting, Group 2: rash, Group 3: myalgia and/or arthralgia, Group 4: headache and/or retro-orbital pain, Group 5: petechiae and/or positive tourniquet test, and Group 6: leukopenia;(b)Probable dengue with warning signs: Any probable case, in addition to fulfilling the dengue clinical picture, presenting one or more of the following warning signs: intense and continuous abdominal pain, persistent or uncontrollable vomiting, fluid accumulation (ascites, pleural or pericardial effusion), mucosal bleeding (epistaxis, gingival bleeding), lethargy or irritability, postural hypotension, hepatomegaly > 2 cm, a progressive increase in hematocrit, platelet count < 100,000 per microliter, or a progressive decrease in platelets and hemoglobin;(c)Probable severe dengue: Any probable dengue case presenting one or more of the following findings: shock due to severe plasma leakage, as evidenced by tachycardia, cold extremities with capillary refill ≥ 3 s, weak or undetectable pulse, convergent differential pressure ≤ 20 mmHg, late-phase hypotension, fluid accumulation leading to respiratory failure, severe bleeding (e.g., hematemesis, melena, massive metrorrhagia, bleeding from the central nervous system), and severe organ involvement such as significant liver damage (AST or ALT > 1000 U/L), kidney involvement, central nervous system (altered consciousness), heart (myocarditis), or other organs.

Laboratory case sampling: All patients meeting the operational case definition of probable dengue were reported to the national epidemiological surveillance system. However, laboratory sampling was sentinel type, with laboratory testing performed for dengue confirmation in 30% of probable non-severe dengue and 100% of dengue cases with warning signs and severe dengue.

Laboratory samples were entered into the IMSS Laboratory Epidemiological Control System, which was the data source for this study, including variables such as age, sex, state of residence, date of symptoms onset, type of care (outpatient or hospitalized), and clinical characteristics of dengue (non-severe dengue, dengue with warning signs, and severe dengue).

The variables studied were sociodemographic (age, sex, state of residence). Age was summarized in years and age groups: under one year, 1–5 years, 6–18 years, 19–40 years, 41–60 years, and 61 years and older. Regarding temporal factors, the date of dengue symptoms onset was used to create disease quarters during the year: January–March, April–June, July–September, and October–December. Regarding geographical location, based on the patient’s state of residence, geographical zones were defined according to the National Institute of Statistics and Geography of Mexico, and the climate of each region (hot and humid, mild and humid, or dry and mild). Hot climate refers to an annual mean temperature greater than 22 °C, mild climate to mean temperatures between 12 °C and 18 °C. Wet climate is defined as regions with year-round or abundant summer rainfall. When these rainfall patterns are not met, it is considered dry.

Regarding clinical characteristics, the patient type was classified as outpatient or hospitalized. According to the operational case definitions of dengue, patients were classified into severe dengue forms based on clinical data from severe dengue cases and dengue with warning signs at the time of clinical care, whether hospitalized or outpatient.

Sample size: All patients with laboratory results for dengue virus serotyping were included, while patients with coinfection by two dengue virus serotypes were excluded (79 laboratory samples).

### Statistical Analysis

A descriptive analysis was conducted to determine the frequency of dengue severity by sex, age group, disease quarter, geographical zone, and dengue virus serotypes. Simple frequencies and proportions were calculated, and a z-test was performed to assess proportional differences between groups. Bivariate analysis with adjusted Chi-square was also conducted. To build the best model, bivariate analysis results were used to construct binary logistic regression models based on the forward Wald method, considering the model’s predictive capacity. The odds ratio was used as a measure of association, with 95% confidence intervals, and statistical significance was set to α < 0.05.

## 3. Results

In the year 2023, a total of 10,441 samples were processed for dengue RT-qPCR at the IMSS, resulting in the following serotype distribution: DENV-3 accounted for 64.4% of the samples (n = 6722), followed by DENV-2 with 22.3% (n = 2333), DENV-1 with 10.7% (n = 1116), and DENV-4 with 2.6% (n = 270). There were more samples from females (52.0%) than males (48.0%). Regarding the condition of care, 63.3% of laboratory samples were from individuals receiving outpatient care, while 36.7% were from hospitalized individuals. The sampling window from symptoms onset ranged from 0 to 5 days, with a median of 3 days and a standard deviation of 1.35 days.

According to clinical classification, 56.7% were reported as non-severe dengue, 41.2% as dengue with warning signs, and only 2.1% were identified as severe dengue.

Table 1 displays population frequencies according to disease severity. There was no significant difference in gender distribution among the different severity groups. However, there was significant variation in age group distribution among dengue cases with different severity grades. Hospitalization rates significantly increased with severity. The distribution by climatic region varied significantly among severity groups. Warmer regions showed higher proportions of dengue cases overall and significant proportions of cases with warning signs and severe dengue.

Cases of dengue were more frequent from July to September. The distribution of dengue virus serotypes varied during the year, with DENV-3 being the more common dengue (Figure 1).

In the bivariate analysis (Table 2), gender was not associated with severity (OR = 0.94, 95% CI: 0.87; 1.02), while extremes of age, especially over 65 years (OR = 2.16, 95% CI: 1.74; 2.70), were associated with higher risk. Warmer regions showed a higher risk (OR = 1.28, 95% CI: 1.11; 1.48), as did the third quarter of the year from July to September (OR = 1.20, 95% CI: 1.10; 1.30). Dengue virus serotype was associated with severity, with DENV 2 (OR = 1.25, 95% CI: 1.14; 1.34) and DENV 4 (OR = 0.68, 95% CI: 0.53; 0.88) showing significant associations.

Eight binary logistic regression models were performed, with Table 3 showing the model with the highest predictive percentage for severe dengue. Women had a higher risk of developing severe dengue than men (OR = 1.086, 95% CI: 1.004; 1.175). Age also influenced the risk of severe dengue, with those under one year and over 65 years having a higher risk at OR = 1.78, 95% CI: 1.004; 3.15 and OR = 1.83, 95% CI: 1.45; 2.31, respectively.

DENV serotypes had different impacts on the risk of severe dengue, with DENV-2 being more likely to cause severe dengue than other serotypes (OR = 1.185, 95% CI: 1.024; 1.371) and DENV-4 being less likely (OR = 0.696, 95% CI: 0.596; 0.922). DENV-3 showed no association with dengue severity.

Quarterly, the models did not show statistical significance for disease and climate regions. The binary regression model had an overall predictive value of 58.5%, with a statistical significance of *p* = 0.000.

## 4. Discussion

Severe dengue analysis was conducted in a population with social security in Mexico, where the continuous introduction and rise of DENV-3 was observed in the southern and southeastern regions of the country in 2023.

There is controversy regarding the relationship between sex and disease development. Some studies found that sex was not related to dengue severity [14,15]. However, in the studied population, women were associated with severe forms of dengue, consistent with other studies [16].

Although dengue was more frequent in the second half of the year in the northern hemisphere, no quarterly association was found with severe forms of dengue.

No statistically significant association was found between severe dengue forms and regions by climate, suggesting that the frequency of severe forms is equal across all regions. However, 91.7% of the analyzed samples originated from areas with warm and humid climates. The average daily temperature and temperature variation have been described as two of the most important drivers of the current distribution and incidence of dengue [17].

Dengue severity has been associated with different serotypes, depending on the region of the world [18]. In our study, DENV2 was strongly associated with dengue severity, while DENV4 remained a protective factor compared to all serotypes.

One limitation of this study was the absence of variables related to clinical conditions and medical history such as pre-existing diseases. However, the strengths of the study include (1) being conducted in a population with social security covering a large part of the national territory; (2) using samples from cases reported in the epidemiological surveillance system; and (3) confirming infection through RT-qPCR and determining the serotype.

This study is one of the first of its kind conducted in a population with social security in Mexico. This analysis provides insights into severe dengue within a population covered by social security in Mexico. It observes the continuous introduction and rise of DENV-3, particularly in Mexico’s southern and southeastern regions during 2023. This study discusses the debate surrounding the relationship between sex and dengue severity, noting discrepancies across different study findings.

Additionally, the absence of a significant association between severe dengue forms and regions by climate suggests a uniform distribution of severe cases across different climatic zones, with most originating in warm and humid areas. This observation underscores the influence of temperature on dengue incidence and distribution, consistent with previous literature.

Dengue severity was strongly associated with DENV2 in this study, as opposed to DENV4, which appears as a protective factor compared to all serotypes. These findings contribute to our understanding of the complex interplay between serotypes and disease severity, offering insights for public health interventions and further research.

## Figures and Tables

**Figure 1 viruses-16-00769-f001:**
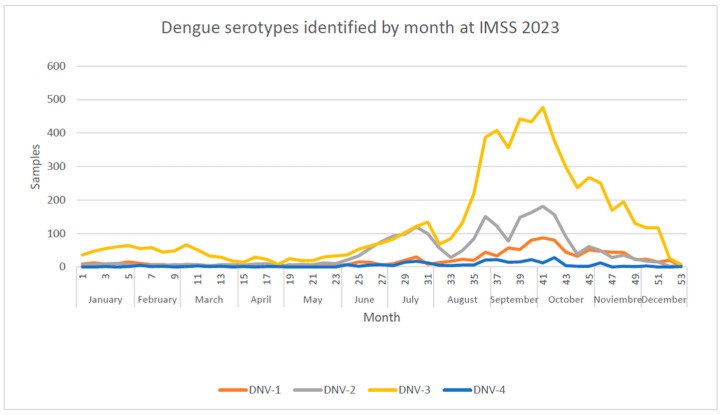
Dengue serotypes identified in 2023 by week at IMSS 2023.

**Table 1 viruses-16-00769-t001:** Clinical epidemiological characteristics for dengue severity.

Variables		Total	Dengue with Wartning Signs and Severe Dengue				*p* *
			NOT		YES		
		n	n	%	n	%	
Clinical classification	Dengue without Signs	5922	5922	100.0%	0	0.0%	<0.05 *
	Dengue with Warning Signs	4300	0	0.0%	4300	95.2%	<0.05 *
	Severe Dengue	219	0	0.0%	219	4.8%	<0.05 *
Gender	Female	5433	3046	51.4%	2387	52.8%	>0.05
	Male	5008	2876	48.6%	2132	47.2%	>0.05
	Under 1 year old	50	20	0.3%	30	0.7%	<0.05 *
Age group	1 to 4 years	172	115	1.9%	57	1.3%	<0.05 *
	5 to 14 years	2520	1222	20.6%	1298	28.7%	<0.05 *
	15 to 24 years	2453	1346	22.7%	1107	24.5%	<0.05 *
	25 to 44 years	3802	2555	38.1%	1247	27.6%	<0.05 *
	45 to 64 years	1397	831	14.0%	566	12.5%	<0.05 *
	65 and older	347	133	2.2%	214	4.7%	<0.05 *
Hospitalized	NO	6609	5567	94.0%	1042	23.1%	<0.05 *
	YES	3832	355	6.0%	3477	76.9%	<0.05 *
Region by climate	Warm humid and sub-humid	9574	5383	90.9%	4191	92.7%	<0.05 *
	Temperate humid and sub-humid	611	384	6.5%	227	5.0%	<0.05 *
	Very dry and dry	256	155	2.6%	101	2.2%	>0.05
Trimester of signs and symptoms onset	January–February–March	847	616	10.4%	231	5.1%	<0.05 *
	April–May–June	680	427	7.2%	253	5.6%	<0.05 *
	July–August–September	4497	2363	39.9%	2134	47.2%	<0.05 *
	October–November–December	4417	2516	42.5%	1901	42.1%	>0.05
DENV1	NO	9325	5299	89.5%	4026	89.1%	>0.05
	YES	1116	623	10.5%	493	10.9%	>0.05
DENV2	NO	8108	4699	79.3%	3409	75.4%	<0.05 *
	YES	2333	1223	20.7%	1110	24.6%	<0.05 *
DENV3	NO	3719	2023	34.2%	1696	37.5%	<0.05 *
	YES	6722	3899	65.8%	2823	62.5%	<0.05 *
DENV4	NO	10,160	5736	97.0%	4424	97.9%	<0.05 *
	YES	270	177	3.0%	93	2.1%	<0.05 *

* z-test.

**Table 2 viruses-16-00769-t002:** Bivariate analysis of severe dengue risk factors.

Variables		Odds Ratio	IC 95%		*p*
			Lower	Upper	
Gender	Female	0.946	0.875	1.022	0.16
Age group	Under 1 year old	1.97	1.12	3.48	0.017
	1 to 4 years	0.65	0.47	0.89	0.007
	5 to 14 years	1.55	1.42	1.70	0.000
	15 to 24 years	1.10	1.00	1.21	0.035
	25 to 44 years	0.62	0.57	0.67	0.000
	45 to 64 years	0.88	0.78	0.98	0.025
	65 and older	2.16	1.74	2.70	0.000
Region by climate	Temperate humid and sub-humid	0.76	0.64	0.90	0.002
	Warm humid and sub-humid	1.28	1.11	1.48	0.001
	Very dry and dry	0.85	0.66	1.10	0.211
Trimester of signs and symptoms onset	January–February–March	0.50	0.42	0.58	0.000
	April–May–June	0.78	0.66	0.93	0.004
	July–August–September	1.20	1.10	1.30	0.000
	October–November–December	1.00			
Serotype DENV	DENV1	1.04	0.92	1.18	0.523
	DENV2	1.25	1.14	1.37	0.000
	DENV3	0.86	0.80	0.94	0.000
	DENV4	0.68	0.53	0.88	0.003

**Table 3 viruses-16-00769-t003:** Logistic regression analysis of severe dengue risk factors.

Variables		Odds Ratio	CI 95%		*p*
			Lower	Upper	
Gender	Woman	1.086	1.004	1.175	0.040
Age group	Under 1 year old	1.780	1.004	3.156	0.048
	1 to 4 years	0.590	0.425	0.819	0.002
	5 to 14 years	1.286	1.150	1.438	0.000
	25 to 44 years	0.660	0.593	0.733	0.000
	45 to 64 years	0.805	0.704	0.920	0.001
	65 and older	1.838	1.456	2.318	0.000
	DENV-1	1.000			
Serotype DENV	DENV-2	1.185	1.024	1.371	0.023
	DENV-3	0.938	0.824	1.068	0.333
	DENV-4	0.696	0.526	0.922	0.011
Constant		−0.213			0.004

## Data Availability

All the data is available at the following link https://doi.org/10.6084/m9.figshare.25267402.v1. Created on 1 February 2024.

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
