# Peer review of "Retrospective Analysis of Severe Dengue by Dengue Virus Serotypes in a Population with Social Security, Mexico 2023"

_viruses, 2024, doi:10.3390/v16050769_

Round 1

Reviewer 1 Report

Comments and Suggestions for Authors

Manuscript reference: Viruses-2939035

Title:  Epidemiological characterization of dengue virus phenotypes in Mexico: implications for disease severity

Article Type: article

General comment

Characterizing the epidemiology and the impact of newly circulating dengue virus (DENV) serotypes is of paramount importance for adapting the public health response during outbreaks or surges in dengue endemic countries, which may differ according to the pathogen virulence (genotypically-driven fitness to the vector for developing high viral loads and subsequent  high circulating viremia, virus-specific factors), or the background immunity and other characteristics of the host population (age, comorbidities, genetics). In this framework, Hernandez-Bautista and coworkers provide extensive data on the 2022-2023 DENV3 epidemic in Mexico, a known hyperendemic country for dengue transmission, previously affected by recurrent DENV-2 and DENV-1 outbreaks. The amount of work done is important and the article has the potential for interesting a broad readership, however there are several major issues to solve before it can be accepted, including the working hypotheses (or the options chosen for the analysis), the organization of the data, the analysis, the interpretation, and the reporting of study findings. I highly recommend the help of a professional statistician to improve the description of the methods and explanations of results, given the too many difficulties to understand what was really done. I also recommend the help of an English fluent proofreader to improve the quality of the manuscript, given too many imprecisions and ambiguities. Last, I recommend the use of STROBE guidelines for better organizing the paper and making it more readable without requiring salicylates or other pain killers to prevent the headaches.

Specific comments

Title and abstract

1(a). Indicate the study’s design with a commonly used term in the title or the abstract. I would indicate that the study consists in a retrospective analysis of epidemiologic surveillance data based on a national laboratory network.

1(b). Provide in the abstract an informative and balanced summary of what was done and what was found. This is far to be the case in the current version, the presented findings
seeming to respond to an arbitrary selection rather than a very structured plan. The abstract should also specify that the surveillance is exclusively virologic and based on RT-qPCR confirmed cases and that it does not include cases diagnosed using rapid tests (RDTs), serology, nor possible cases using the syndromic definition of probable dengue without testing.

Page 1, line 33. Write “Despite, its low prevalence…” even though the term prevalence does not fit its definition, from an epidemiologic point of view.

Page 2, lines 30 and 33. The stronger association between DENV-2 and hospitalization seems relying on a comparison taking DENV-4 as reference serotype.

Introduction

2. Background/rationale

Explain the scientific background and rationale for the investigation being reported. This section is a bit long and include some redundancies. This should be shortened for better fluidity and clarity. The paragraph on dengue serotypes historical circulation and those depicting the current situation regarding the DENV3 surge in the Americas, central America and Mexico should be reordered in a logical manner.

Page 1, line 40. Define what is a hyperendemic area for dengue or give your frank personal arguments for thinking Mexico is hyperendemic and not merely endemic. This will help the readers to understand the epidemiologic situation.

Page 2, lines 68 and 69. Predominance of DENV-1: 2011 to 2018, or 2016 to 2018? Better write that DENV-1 dominated in 2011-2017 and DENV-2 in 2019-2022, and that 2018 was a year of codominance or cocirculation, also specifying that during those years, DENV-3 and DENV-4 were also present.

Page 2, lines 77 to 79. Specify the window of opportunity recommended for each test by the SISCEP or the Mexican Ministry of Health. The choice of the test depends on the time to presentation post symptom onset. For instance, the window to better obtain a positive test is as follows: early acute phase (day 0 to day 5 post symptom onset) for RT-qPCR and RDT alone or combined with IgG antibodies detection (early ascension indicating secondary dengue), late acute phase (day 5 to day 7) for RT-qPCR and IgM-IgG antibodies (for distinction between primary and secondary dengue) and convalescence phase for IgM-IgG serology.

3. Objectives

Objectives are ill characterized. State specific objectives, including any prespecified hypotheses. This can be done specifying a primary objective, for instance, assessing the impact of dengue on dengue outcomes (hospitalization, length of stay and dengue-related deaths) and secondary outcomes (dengue classification, serotype-associated characteristics).

Page 2, lines 81 to 83. Provide a reference on DENV3-associated severity or reformulate the sentence as a hypothesis you want to investigate. Each objective should be supported by hypotheses. The hypotheses should be ordered in a logical manner. The article is very rich and detains information on dengue incidence, dengue prevalence, dengue presentation and outcomes. The objectives should be investigated as follows: the incidence first and its regional variations, the prevalence (as the result of incidences in the 2022-2023 outbreak and former epidemics) and its regional variations, the clinical presentation (dengue fever without warning signs [DFnWS], dengue fever with warning signs [DFWS] and (severe dengue [SD]) and outcomes (hospitalization, length of stay and death. This will condition the order of apparition of the tables with descriptions followed by investigation risk factors or prognostic factors.

Methods

4. Study design

Present key elements of study design early in the paper. This should specify that the study is a retrospective analysis of epidemiologic surveillance data based on a national laboratory network.

5. Setting

Describe the setting, locations, and relevant dates, including periods of recruitment, exposure, and data collection.

6. Participants

6 (a). Cross-sectional study—Give the eligibility criteria, and the sources and methods of selection of participants. There is no mention of date of presentation post symptom onset. This will help to know if RT-qPCR were performed in the appropriate window of positivity and if differs according to the serotype (the timing of presentation negatively correlates with the perceived degree of severity: the most severe is perceived the deceive, the faster the patient consults and is sampled for RT-qPCR). Was this data collected? Were serologies?

7. Variables

Clearly define all outcomes, exposures, predictors, potential confounders, and effect modifiers. Give diagnostic criteria, if applicable

8. Data sources/ measurement

For each variable of interest, give sources of data and details of methods of assessment (measurement). Describe comparability of assessment methods if there is more than one group

9. Bias

Describe any efforts to address potential sources of bias

10. Study size

Not appropriate here given the retrospective character of the study with no prespecified hypothesis

11. Quantitative variables

Explain how quantitative variables were handled in the analyses. If applicable, describe which groupings were chosen and why. I recommend presenting Ct values as means and 95%CI rather than medians and extreme values, given means are more sensitive to changes and able to show clinically relevant differences. For a clinician, the relevance of the difference between a median Ct of 23.895 and 26.565 is not evident. In addition, analysing the Ct value (or inversely the DENV circulating viremia) makes no sense without controlling the timing of presentation. Ct values may be higher (viremia lower) just because patients present later.

12. Statistical methods.

This chapter should be rewritten by a professional statistician. From the 6 lines dedicated to explain statistical analysis, we do not really understand how multivariate models were constructed. What was the guiding strategy ? How the options were chosen? For instance, it seems irrelevant to choose DENV4 as a reference category as it is caused no epidemic and seldom found in comparison to other serotypes. I rather think each serotype should be investigated using all other serotypes as comparator, and that DENV3 should be also studied in additional sensitivity analyses excluding certain serotypes. For instance, if a serotype different than DENV3 is associated with severe dengue when comparing this serotype against all other serotypes, the relationship between DENV3 and severe dengue should be also analysed once excluded this serotype to know have more chances to identify the risk of severity with DENV3. Idem for the relationship between DENV serotypes and the ability develop warning signs.

12 (a). Describe all statistical methods, including those used to control for confounding. All models should be accompanied by the full bivariate analyses as supplementary tables.

I have checked the statistical analyses whenever the raw data allowed bivariate analyses to be performed using direct commands in Stata®16 and found many errors of reporting, mostly based on inappropriate tests.

12 (b). Describe any methods used to examine subgroups and interactions

12 (c). Explain how missing data were addressed

12 (e). Describe any sensitivity analyses

Results

13. Participants

13 (a). Report numbers of individuals at each stage of study—eg numbers potentially eligible, examined for eligibility, confirmed eligible, included in the study, and analysed

13 (b). Give reasons for non-participation at each stage

13 (c). Consider use of a flow diagram as figure 3 (placed just before the analyses).

14. Descriptive data

14 (a). Give characteristics of study participants (eg demographic, clinical, social) and information on exposures and potential confounders.

14 (b). Indicate number of participants with missing data for each variable of interest

The order of tables and figures should be, as follows:

Figure 1.

Figure 2

Table 1 should be former Table A1 should be completed with all relevant data to subsequent analyses and placed in the text as the new descriptive long Table 1 with no statistics. In the former Table 1, row percentages should have been preferred to column percentages and Fisher’s exact test should have been used wherever chi2 test did not apply.

Table 2 should be former Table 5

Following this supplemental tables should be as follows.

Table A1 should be former Table A3 to present early in the text the regional differences in serotype and coinfection distributions.

Table A2 should be former Table A4 and mirror Table 2

Table A3 should be former Table 1.

Table A4 should be former table A5.

Other tables should be removed, or some elements placed in the new table.

Page 4, line 148. “There was a slightly higher numbers from females with no differences observed in circulating serotype by sex.” This statement is misleading given significant sex differences in the prevalence of DENV-1 and DENV-2 (older Table 1, now Table A3), the first being more common in males (p=0.035), the second more common in females (p=0.003).

Page 4, line 160. How can DENV-2 dominate in hospital management while there is a  threefold more cases of symptomatic dengue

15. Outcome data

Table 3 should be former Table A3 and investigated in bivariate analysis the relationship between dengue serotypes and clinical presentation of dengue. This should be done using all other serotypes (as referenced category) and presenting DENV-4 as the reference country. Row percentages should be presented instead of column percentages.

Table 5 should be former Table 2.

This should investigate in bivariate analysis the relationship between each serotype or coinfection and hospitalization, length of stay, or death vs all other serotypes or coinfection as reference category. Row percentages should also be presented instead of column percentages.

16. Main results

The study of the relationships between dengue serotypes and dengue classification or dengue outcomes in bivariate and multivariate analyses.

This should investigate the relationship between each serotype or coinfection and the clinical presentation vs all other serotypes or coinfection as reference category.

Table 4 should present a multivariate analysis with dengue clinical presentation as the outcome or dependent variable. Multinomial logistic regression with dengue fever with no warning sign [DFnWS] as referent category (Y0), dengue fever with warning sign [DFWS] as Y1 and severe dengue [SD] as Y3.  In exposure, the choice of DENV-4 as reference category should be justified given it has the highest prevalence of SD cases (2.96%).

How DENV1, DENV2 or DENV3 could be associated with increased risk severe dengue, while they all displayed OR <=1 in bivariate analysis (statistical check and Excel edited.

Note that none specific serotype was associated with SD when taking DENV-4 as the reference, and that only DENV-1 was associated with SD when all other serotypes were considered.

Table 6 should present a multivariate analysis with hospitalization and length of stay as outcomes. Also, if ICU stays were known, this would have been interesting to investigators the factors associated with ICU stay or a composite of ICU stay and death.

Older table 4. What was the outcome (or dependant variable) ?

How DENV-4 can be linked to both dengue infection and dengue severity.

Note the term “dengue infection” is inappropriate as dengue is the disease and infections usually refer to a pathogen.

16 (a). Give unadjusted estimates and, if applicable, confounder-adjusted estimates and their precision (eg, 95% confidence interval). Make clear which confounders were adjusted for and why they were included. Unadjusted estimates (crude odds ratios and 95%CI can be placed in the text or as additional supplemental tables. Exp(B) is the OR.

16 (b). Report category boundaries when continuous variables were categorized

16 (c). If relevant, consider translating estimates of relative risk into absolute risk for a meaningful time period

17 Other analyses

Report other analyses done—eg analyses of subgroups and interactions, and sensitivity analyses.  Outcomes should be compared using a specific serotype vs all other serotypes.

Discussion

The discussion is unstructured, which makes the manuscript very hard to evaluate.

18. Key results

Summarise key results with reference to study objectives

Page 11, lines 302. Both DENV1 and DENV2 were associated with sex differences. Revise.

19. Limitations

Discuss limitations of the study, taking into account sources of potential bias or imprecision. Discuss both direction and magnitude of any potential bias

This is not done. The authors present a list of limitations and strength without arguing about their impact on study findings.

20. Interpretation

Give a cautious overall interpretation of results considering objectives, limitations, multiplicity of analyses, results from similar studies, and other relevant evidence.

The literature published in the field is pretty abundant and the authors should discuss it more in details.

21. Generalisability

Discuss the generalisability (external validity) of the study results. The authors should conclude with take home messages and perspectives. It is not sure the data could be extrapolated, given DENV-3 is rarely the predominant pathogen.

Other information

22. Funding

Give the source of funding and the role of the funders for the present study and, if applicable, for the original study on which the present article is based

Additional comments

Place [brackets] in the sentences and not after the points.

Comments on the Quality of English Language

The are redundancies and imprecisions. The paper should be improve by someone fluent in English

Author Response

Dear Reviewer,

We appreciate your comments and corrections to the manuscript. We have addressed each of your observations as detailed below:

The title has been changed to "Retrospective Analysis of Severe Dengue by Dengue Virus Serotypes in a Population with Social Security. Mexico 2023," emphasizing the study's purpose using a studio design. The abstract is now more balanced, indicating what was done and the results found. We have specified that only the laboratory technique by RT-qPCR was used. The background section is shorter, with better fluidity and clarity. We have defined what constitutes a hyperendemic area. We have followed the STORBE guidelines. We find that the manuscript is much clearer now that we have removed the CTs and coinfections from the study. Overall, it is a new document with logic in its procedures and in its reading.

Sincerely, The Authors.

Reviewer 2 Report

Comments and Suggestions for Authors

This is an interesting analysis of positive dengue cases over time in Mexico with associations and implications for disease severity. A few comments:

lines 147-148 The text states there are no differences by sex but the cited Table 1 shows P<0.05 for DNV-1 and DNV-2

lines 290-297 Risk for severe disease is highest in the central region even though rates of secondary infection are higher in other regions. Can the authors speculate on why this may be? Are there just higher numbers of cases in the central region?

lines 315-316 Higher Ct scores in severe cases could also be due to the fact that severe symptoms appear when viral titer is waning, but it would depend on when the test samples were taken from patients that developed severe disease.

Figure 1 - What is meant by 'Week of Symptom Onset'? Is it time from first case sample? To increase legibility, it would be good to reduce the numbers on the x-axis, perhaps an interval of five weeks. Also, under the number of weeks of symptom onset, months of the year these correspond to would be helpful since the authors refer to cases throughout months of the year.

Figure 2 - What do the size of the balls mean? If it corresponds to the number of samples this is indicated on the y-axis and the balls could just be converted to dots for clarity. Again, Increasing the interval of the scale on the x-axis would improve legibility.

Table 3 - Exp(B) should be defined

Table 5 - At the foot of the table 'a' and 'b' designations are defined but do not appear anywhere in the table.

Comments on the Quality of English Language

It would be good to have a native English speaker review the manuscript. There are some minor grammatical changes that would improve readability.

Author Response

Dear Reviewer,

We appreciate your comments and corrections to the manuscript. We have addressed each of your observations as detailed below:

We have reviewed and changed the tables presented. We have not analyzed secondary dengue infections, so we have clarified the manuscript's wording. The week of symptom onset refers to the date the patient started experiencing symptoms, regardless of the date the laboratory sample was taken. In Figure 1, we have changed the x-axis to months and expanded the interval of weeks. We have removed the analysis of CTs from the manuscript to improve the study's comprehension and its objectives. The expression Exp(B) has been defined.

Sincerely, The Authors.
